# Genotype Combinations Drive Variability in the Microbiome Configuration of the Rhizosphere of Maize/Bean Intercropping System

**DOI:** 10.3390/ijms25021288

**Published:** 2024-01-20

**Authors:** Giovanna Lanzavecchia, Giulia Frascarelli, Lorenzo Rocchetti, Elisa Bellucci, Elena Bitocchi, Valerio Di Vittori, Fabiano Sillo, Irene Ferraris, Giada Carta, Massimo Delledonne, Laura Nanni, Roberto Papa

**Affiliations:** 1Department of Agricultural, Food and Environmental Sciences, Marche Polytechnic University, 60131 Ancona, Italy; g.lanzavecchia@staff.univpm.it (G.L.); g.frascarelli@pm.univpm.it (G.F.); lorenzo.rocchetti@staff.univpm.it (L.R.); e.bellucci@staff.univpm.it (E.B.); e.bitocchi@staff.univpm.it (E.B.); v.divittori@staff.univpm.it (V.D.V.); 2National Research Council of Italy, Institute for Sustainable Plant, Strada delle Cacce 73, 10135 Torino, Italy; fabiano.sillo@ipsp.cnr.it; 3Department of Biotechnologies, Strada le Grazie 15, 37134 Verona, Italy; irene.ferraris@univr.it (I.F.); giada.carta@univr.it (G.C.); massimo.delledonne@univr.it (M.D.)

**Keywords:** *Zea mays* L., *Phaseolus* spp., high throughput sequencing, 16S region, bacterial community, metagenomics, rhizosphere, intercropping

## Abstract

In an intercropping system, the interplay between cereals and legumes, which is strongly driven by the complementarity of below-ground structures and their interactions with the soil microbiome, raises a fundamental query: Can different genotypes alter the configuration of the rhizosphere microbial communities? To address this issue, we conducted a field study, probing the effects of intercropping and diverse maize (*Zea mays* L.) and bean (*Phaseolus vulgaris* L., *Phaseolus coccineus* L.) genotype combinations. Through amplicon sequencing of bacterial 16S rRNA genes from rhizosphere samples, our results unveil that the intercropping condition alters the rhizosphere bacterial communities, but that the degree of this impact is substantially affected by specific genotype combinations. Overall, intercropping allows the recruitment of exclusive bacterial species and enhances community complexity. Nevertheless, combinations of maize and bean genotypes determine two distinct groups characterized by higher or lower bacterial community diversity and complexity, which are influenced by the specific bean line associated. Moreover, intercropped maize lines exhibit varying propensities in recruiting bacterial members with more responsive lines showing preferential interactions with specific microorganisms. Our study conclusively shows that genotype has an impact on the rhizosphere microbiome and that a careful selection of genotype combinations for both species involved is essential to achieve compatibility optimization in intercropping.

## 1. Introduction

Intercropping, i.e., the simultaneous cultivation of different plant species on the same field is a promising way to diversify crops [1] and to promote sustainable agriculture [2,3]. This approach offers multiple ecological services [4,5,6], including enriched biodiversity [7], improved resource efficiency and enhanced soil health [8]. It optimizes nutrient cycling, conserves water, and increases resilience to environmental challenges. Evidence also indicates a positive correlation between greater plant diversity and increased microbial abundance, diversity, and soil carbon sequestration [9,10].

In particular, legume intercropping is a promising way to eco-intensify the production of cropping systems [4,5,6] and can be used to enrich soil with biologically fixed N due to legume–rhizobia symbiosis, thereby improving the soil quality [11]. In addition, the mixture of two or more crop species like cereals and legumes with different root systems and rhizosphere activities often generates better cover and efficient soil exploration for better resource uptake [12]. However, challenges in promoting intercropping adoption exist. These challenges include scarce breeding efforts and limited availability of genomic tools. Such tools are essential for optimizing legume adaptation to intercropping, dissecting the dynamics of the interaction, and understanding their effects on the different genotype combinations used in the intercropping system [13,14]. The selection of varieties that can efficiently exploit the benefits of co-cultivation remains a technical limit to dissemination in agricultural systems [15]. 

Recent studies highlight that society seeks sustainable agroecosystems and improved provision of ecosystem services, even if it results in higher prices for fruits and vegetables. It is proposed that a holistic approach to exploit crop diversification may represent an efficient alternative to intensive monocropping. Indeed, farmers and society recognize crop diversification as an effective method to build the resilience of agro-ecosystems through adaptive management of input factors and environmental challenges. As a result, it is suggested that environmentally oriented policies would be largely embraced [16]. However, to encourage the adoption of intercropping, it is imperative to develop indicators to measure the delivery of ecological services. Typically, benefits are assessed using conventional metrics like crop yield or resource-use efficiency or reduction in the use of agro-chemicals, but less attention has been given to mechanisms involved in soil health. Nevertheless, belowground interactions play a crucial role in shaping facilitative processes [11,17,18,19,20,21]. Intercropping favors the development of different types of roots and changes overall root distribution and architecture, as well as exudation processes in the rhizosphere [22]. Consequently, intercropping will influence both the extent and nature of the relationships between plants and microorganisms, thereby enabling new beneficial interactions [1]. More broadly, it has been postulated that the rhizosphere microbiota may underlie the added value of intercropping [22]. Evidence suggests that the influence of root systems on microbial communities, known as the “rhizosphere effect”, is exerted both at species that at genotype level, as observed in maize, soybean, and alfalfa [23,24,25,26,27,28]. The species-specific effect has been observed in intercropping systems as well. Indeed, various studies on cereal/legume intercropping systems have documented shifts in microbial communities influenced by the choice of co-cultivated species [29,30,31]. These studies have observed changes in the abundance of bacteria belonging to specific phyla, with some increasing or decreasing either in cereals or legumes. However, even when these studies shared the same plant species combined in intercropping, the reported microbial changes exhibited deep variability from one study to another. This suggests that, in addition to crop species, the choice of crop genotypes may play a role in influencing microbial composition, further supporting the idea of a genotype-specific effect on rhizosphere communities. Furthermore, recent findings highlighting different responses in the root bacterial communities of two sugarcane varieties intercropped with soybean [32] and variations in community structures between different varieties of wheat and pea in intercropping [33] further validate the hypothesis of a genotype-driven impact on the microbiome in intercropping scenarios.

These studies emphasize the challenge of establishing universal relationships between plants and microorganisms and the increased complexity of characterizing microbial communities in intercropping systems [34]. Given the highly specific nature of plant–soil microorganism relationships, our study, which aims to investigate the maize (*Zea mays* L.)/bean (*Phaseolus vulgaris* L. and *Phaseolus coccineus* L.) intercropping system, incorporates two primary considerations based on the aforementioned observations. Firstly, observing microbiome variations in intercropping by focusing on individual plant lines may yield limited or overly specific insights. Conversely, identifying common patterns across multiple varieties grown in intercropping can reveal more reliable trends. Secondly, examining microbiome profiles in different intercropping combinations can determine whether a universal trend exists or if it varies with the selection of co-cultivated genotypes, shedding light on genotype-specific effects on microbial recruitment and their direct involvement in the intercropping system. For this scope, we applied a comprehensive blind approach to identify, at first stance, the overall impact of intercropping (IC) condition on the rhizosphere microbiome compared to sole crop (SC) and, secondly, to characterize and compare the impact of various maize and bean lines (varieties) combinations in IC. Diversity, composition, and structure of bacterial communities were characterized by high throughput sequencing of 16S rRNA genes and results are discussed first in the impact of IC as a whole and then of various maize/bean genotypes combinations in IC. 

## 2. Results

### 2.1. Composition of the Bacterial Community in the Rhizosphere of Maize/Bean Intercropping System

Plant status (maize in SC; beans in SC; maize/bean IC) has a significant effect on Chao1 and Shannon richness and evenness diversity indexes (*p* value < 0.05) (Appendix A). Beans in SC show a significantly lower bacterial diversity compared to maize in SC, while the intercropping condition shows an intermediate behavior between maize and beans in SC (Figure 1a). Means of Shannon indexes for plant status have similar dynamics (Appendix A), suggesting that both the richness and evenness of bacterial communities are impacted by the IC condition. Concentrating on biologically relevant ASVs from trimmed and filtrated tables (3134 taxa), the rhizosphere in intercropping condition includes 508 private species (Figure 1b) for a total of 10,334 reads mostly belonging in terms of diversity and abundances to the phyla Proteobacteria (27% and 32%, respectively) (Appendix A). Exploring private bacterial species in the IC rhizosphere and concentrating on higher taxonomic levels (order) for simpler interpretation, some bacterial orders (21) are identified as being absent in maize in SC, such as the order Rokubacteriales and Latescibacterales. Others are present only in the intersection between maize in IC and beans, as Methylococcales and Nitrosococcales along with other bacterial orders absent in maize in SC and that are retrieved from the beans’ rhizosphere as Acidimicrobiales, Methylacidiphilales and Fibrobacterales. 

The rhizosphere of IC condition is mainly constituted by Proteobacteria (31%) with the orders Betaproteobacteriales, Myxococcales and Betaproteobacteriales (Figure 1c) enclosing the highest diversity within these phyla. Rhizosphere in IC shows an increment in relative abundances of Actinobacteria and Entotheonellaeota compared to maize and beans in SC, while Proteobacteria show an increment in IC compared to maize in SC but relative abundances are comparable to the one of beans in SC. Gemmatimonadetes and Acidobacteria have slightly lower abundances compared to SC (Figure 1a). The most abundant phylum in intercropping is Actinobacteria (40%) (Figure 2b) which is also identified as a biomarker for the intercropping system by LefSe analysis (Figure 2c,d, Appendix A). Within Actinobacteria, the orders Propionibacteriales, Streptomycetales, Micrococcales and Corynebacteriales are reported as a biomarker of the IC system, thus they show higher relative abundances compared to SC conditions. At the phylum level, Entotheonellaeota also results as a biomarker of IC condition.

### 2.2. Differentiation of the Bacterial Community Structure in the Rhizosphere of the Maize/Bean Intercropping System Compared to Sole Crop

Properties of correlation networks built for maize in SC and for intercropping conditions highlight that network complexity increases in IC (IC: 866 nodes, 48,783 edges, 0.13 density) compared to maize in SC (Maize_SC: 626 nodes, 2488 edges, 0.0126 density) with an increment of significant edges of ~20 folds and being 10 times denser than the SC network. On the other side, correlation networks built for beans in SC highlight a community structure characterized by higher network complexity than the one showed by maize in SC and nearer to the one observed in the IC condition (Bean_SC: 647 noded, 28,939 edges, 0.0872 density). Network differential analysis conducted to compare correlation networks built for the SC condition of each species (maize and bean) against the IC condition highlighted that the bean in SC does not show significantly differentially correlated nodes compared to the IC condition (verifying the resemblance in structure noted with network properties.). On the other hand, differentially associated nodes are observed for the comparison between maize in SC against the IC condition. Indeed, correlation networks (Figure 3) built with only differentially associated taxa between maize in SC and IC show a higher number of connections in IC compared to SC, reducing the number of clusters (Figure 3) between members of the bacterial community in IC. In particular, among differentially associated taxa in the IC network, the one with the highest number of connections is the order Rhizobiales (97 nodes). A significant number of connections are shown also for the Chloroflexi_Ellin 6543 and the order Frankiales. The order Myxococcales shows the highest increment of connections (40 folds) in IC. The majority of nodes highlighted for being differentially connected in the two networks mostly belong to the phyla Proteobacteria and Actinobacteria, confirming the pivotal role of these two phyla in shaping the IC condition (Table 1).

### 2.3. Composition of Rhizosphere Bacterial Community in Different Maize/Bean Intercropping Combinations

Bacterial community richness (Chao1 index) is significantly impacted by different IC combinations (“Accession”) within each “Environment” (bean lines in SC, maize lines in SC and respective IC combinations) (*p* < 0.005) (Appendix A). Shannon index, instead, is not significantly impacted by different IC combinations. Within bean lines, pC shows a significantly lower richness compared to the other lines of *P. vulgaris*. The two lines p83 and p91 have lower richness compared to pM and pMNE2. Bacterial richness shows significant differences among the maize/bean combinations in the IC systems determined by the maize landraces BP and O. Means of Chao1 index values for maize lines intercropped with p83 and p91 show similar behavior with a decrement of the bacterial richness, while IC combinations with pM and pMNE2 there is an increment of the bacterial richness. This trend resembles the one observed for the corresponding lines of Phaseolus in SC and is maintained in all the “Environment” except for the one determined by maize line I (Figure 4a).

CPCoA analysis on Bray–Curtis matrices identified a significant differentiation of the bacterial communities determined by the most contributing phyla in the rhizosphere of IC combinations with BP (*p* value < 0.035) and O maize lines (*p* value < 0.012). In both cases, centroids of maize lines in SC (BP, O) clusters separately from their IC combinations. The space identified by abundances of bacterial taxa belonging to the most contributing phyla highlights a trend for which IC combinations with BP and O are mostly located where there is a condition of higher abundances and diversities of the microbial community. Interestingly, centroids of maize BP intercropped with p83 and p91 are nearly overlapping and centroids of maize O intercropped with p83 and p91 are located in a space with similar bacterial composition, even if the first is dominated by Actinobacteria and the second by Proteobacteria. In general, for IC “Environments” determined by BP and O maize lines, it seems that IC combinations with p83 and p91 tend to cluster in the same bacterial community space, while the behavior of IC combinations with pM and pMNE2 diverges. Possibly, centroids of these combinations fall in a space determined by relative abundances of diverse phyla, which may contribute less to the bacterial community composition of maize/bean intercropping as a whole but which enclose the higher diversity highlighted by the Alpha-diversity index. In all “Environments”, bacterial communities are dominated by the presence of Actinobacteria and Proteobacteria (Figure 4b and Appendix A). Permanova (Table 2) shows that Plant status had no significant effect on Bray–Curtis within each “Environment”, but a significant effect of “Accession” is observed in BP and O IC systems, suggesting that significant differentiation is dependent on specific combinations between maize and bean lines.

The majority of bacterial genera differentially expressed (DE) are downregulated in single IC combinations compared to plants in SC (Appendix A). Down-regulated taxa account for relatively small abundances while up-regulated taxa have higher relative abundances, in particular for IC combinations with BP. Taxa differentially expressed mostly belong to phyla Actinobacteria and Proteobacteria and some recurrent bacterial taxa are up-regulated in different IC combinations compared to their maize counterpart in SC. Among bacterial genera DE that are upregulated in IC compared to SC are especially present members of the phyla Actinobacteria, Proteobacteria, Planctomycetes and Chloroflexi. IC combinations with BP and O are the ones showing the highest number of DE bacterial genera (Figure 5a). BP rhizosphere in intercropping conditions shows enrichment of bacterial genera belonging to the phyla Actinobacteria, while O in intercropping is enriched of bacterial communities belonging to the phyla Proteobacteria (Figure 5b). Bacterial taxa that are recurrently upregulated in different IC combinations with maize lines BP and O are reported in Figure 5c. These members of the bacterial communities may have a pivotal role in shaping the bacterial communities in intercropping conditions.

### 2.4. Bacterial Community Structure in the Rhizosphere of Different Maize/Bean Intercropping Combinations

Co-occurrence networks built for plants grown in SC and for IC combinations highlight that connections among bacterial communities are modified in IC compared to respective maize lines in SC (Figure 6a). The network of bean landrace p91 behaves differentially from pM and pMNE2. The former shows a low number of nodes and edges and higher modularity, while the latter has a high number of nodes and edges and lower modularity which implies increased network complexity and stronger connection among microbial members. The IC networks shift towards structures resembling the ones of the bean line associated; S-pM, S-pMNE2, BP-pMNE2 and O-pM networks show increased complexity compared to the correspondent maize line in SC. Conversely for IC with p91, S-p91, BP-p91, O-p91 and I-p91 networks experience a reduction in the number of nodes and edges, thus lower complexity. Similar behavior is observed for IC with p83 (BP-p83, O-p83, I-p83). Number of connections (degrees) of each node grouped by phyla membership (Figure 6b) ulteriorly shows the divergent network structures of IC combinations with landraces p83 and p91 from IC combinations with pM and pMNE2, confirming the trend of higher number of connections in the former and lower in the latter. Actinobacteria and Proteobacteria are the two phyla showing a stronger increment of connections in the IC combinations with pM and pMNE2. 

## 3. Discussion

Lately, intercropping practice is receiving more attention due to its beneficial environmental effects including on soil fertility and nutrient availability. Among the mechanisms that are relevant for a better understanding of the dynamics involved in the co-cultivation of plant species, the interaction with the microbial members of the soil is of pivotal importance, in particular when considering cereals and legumes [35]. In this regard, studies on rhizobacterial communities involved in intercropping with maize have been conducted for different associations with leguminous crops: i.e., Maize/Soybean [36,37], Maize/Peanut [38,39,40,41,42], Maize/Faba bean [43,44,45,46,47,48], Maize/*Desmodium* [49]. These studies validated a remarkable rearrangement of the rhizosphere microbiome and a significative effect of plant species involved in the intercropping condition [30,50]. However, while the impacts on rhizosphere microbiome driven by different species in maize/legume intercropping systems have been investigated, little has been undertaken to elucidate the effects of combining different lines of the species involved. Recent studies provided evidence of maize genotype-specific exudation strategies, including their implication for microbial functions [51], corroborating the results on the genotype specificity of the “rhizosphere effect” [24]. Given that intercropping is influenced by synergies happening between root systems and that this likely has an impact on the exudation profiles of the plant species involved [1], the present study aims to determine whether the specificity of the plant–microbiome interactions observed in co-cultivation condition can be extended also at the genotype level. For this purpose, our study tries to encompass both the specificity of the “rhizosphere effect”, thus exploring the changes determined by the intercropping (IC) condition compared to sole crop (SC) as a whole, that is, the potential effect determined by different combinations of genotypes. Thus, we compared the rhizosphere bacterial communities of four maize lines (Biancoperla BP, Ottofile O, Spinato S, Ibrido I) and five bean lines (p83, p91, pM, pMNE2, pC) in sole crop (SC) and in co-cultivation, testing all the possible combinations in intercropping (IC). Bacterial communities were characterized and compared for their diversity, composition, and structure through the analysis of 16s sequencing data. Results are discussed in the impact of IC on rhizosphere microbiome compared to maize and bean in SC as a whole and then reporting the changes observed in specific genotypes combinations in IC. 

### 3.1. Members of Proteobacteria and Actinobacteria Phyla Are at the Basis of the Reconfiguration of the Bacterial Community in the Rhizosphere of Maize/Bean Intercropping System

Maize and bean in SC significantly differentiate their bacterial community diversities, confirming the specificity of rhizosphere bacterial recruitment [52,53]. Bacterial richness represented by Chao1 index is higher in maize compared to beans in SC, suggesting a relevant contribution of rare species in the former. However, the evenness of bacterial population based on the Shannon index shows the same behavior, suggesting that not only rare species are more present in maize in SC but that also common ones are more represented and abundant in maize compared to beans. IC, despite not showing statistically significant differences, shifts its bacterial richness towards intermediate values between maize and beans in SC (Figure 1). This may be explained by the fact that, realistically, the IC condition maintains the starting maize microbiome composition which is then influenced by the presence of the co-associated bean. Alternatively, the observed diversity in IC can be ascribed to the average value determined by employing multiple genotypes which capture a portion of the intra-specific variability of the species involved in the IC combinations.

Concentrating on biologically relevant ASVs, total diversity in IC is dominated by the phylum Proteobacteria (31%), which increases the contribution to total diversity compared to maize in SC (28%) along with Actinobacteria (IC = 13%, maize SC = 8%), while Bacteroidetes (IC = 11%, maize SC = 13%) and Acidobacteria (IC = 19%, maize SC = 3%) decrease their contribute. Interestingly, the IC condition enables the recruitment of members of the soil microbiome that were not recruited by maize and beans in SC. Private bacterial species (508) in IC mostly belong to the phyla Proteobacteria followed by Actinobacteria. Grouping the private species individuated by their higher taxonomic levels, it was possible to identify some bacterial orders that are effectively present only in IC (21). Among these, with ecological significance, we highlight the order Rokubacteriales which belong to a newly identified phyla (Rokubacteria) positively correlated with suppression of fungal pathogens [50,54] and the order Latescibacterales that show plant-growth promoting activity (PGPR) and that is involved in C metabolism [55]. Other bacterial orders of the IC condition appeared to be private of the intersection between beans in SC and IC (group arising to 198 species). Among these, Methylococcales, potential plant growth-promoting bacteria (PGPB) known for their methanotrophic activity and ability to synthesize auxins and induce plant morphogenesis [56,57]; Nitrosococcales, having a role in enhancing plant growth, carbon and nitrogen fixation [55]; Acidimicrobiales which Cheng et al., [58] found to be involved in soil nitrification; Methylacidiphilales [59], involved in heterotrophic metabolism and Fibrobacterales [60], which are associated to extraradical hyphae of arbuscular mycorrhizal fungi (AMF). Our results show that in some cases, intercropping introduces microbes to the roots of the other species, and in other cases, some microbes appear in intercropping conditions without necessarily being introduced by the associated species. This further supports the hypothesis that mixed-cropping leads to novel exudates that alter microbial community composition [61].

Despite diversity in IC is dominated by the phylum Proteobacteria, the main contributor in terms of abundance is the phylum Actinobacteria. Actinobacteria were already observed as the most dominant rhizobacterial group in the intercropped legume soil [30] and have been reported for their beneficial interaction with plants [62,63]. Actinobacteria include 40% of the total bacterial abundances in IC and it is identified for being a biomarker of the intercropping condition. It is the highest taxonomic level experiencing a significant change in abundances when compared to maize and bean in SC. Within the phylum Actinobacteria, are identified as biomarkers of the IC condition the bacterial order Propionibacteriales with the family Nocardioidaceae, which have a role as PGPB [64,65], the order Streptomycetales with the family Streptomycetaceae and genus *Streptomyces* which are always involved in mechanisms of plant growth promotion [66], the bacterial order Micrococcales, known for being an AMF-associated bacterial order [67], with Families Intrasporangiaceae and Microbacteriaceae and the order Corynebacteriales. Apart from Actinobacteria, are identified as biomarkers members of the phylum Entotheonellaeota and the family Archangiaceae involved in the carbon cycle within the phylum Proteobacteria [68] (Figure 2). 

The role of Proteobacteria and Actinobacteria members in shaping the new configuration of the rhizosphere microbiome in IC is determinant also for the structure of the microbial network of IC compared to the one of maize in SC. Indeed, the microbial community structure in IC is significantly altered compared to the one of maize in SC and the taxa highlighted for being significantly differentiated mostly belong to the phyla Actinobacteria and Proteobacteria. In particular, seven Actinobacteria and nine Proteobacteria show a strong increment of connections in IC compared to maize in SC and, within differentially expressed taxa in IC, the one showing the highest number of connections in the new configuration is the order Rhizobiales. Significative increments are observed also for the Actinobacteria of order Frankiales, Propionibacteriales, Micromonosporales and Euzebyales but the strongest change in number of connections from SC to IC is recorded by a member of the class Deltaproteobacteria. When examining the network properties of the IC condition in contrast to those in SC, it becomes evident that IC exhibits a heightened complexity in its community structure, especially when compared to maize in SC. Indeed, while bean in SC shows a number of nodes and, above all, a number of connections nearer to the ones of IC, and maize in SC is instead characterized by significantly lower complexity (Figure 3). This may suggest that bean rhizosphere strongly influences one of the IC systems but that, at the same time, the IC condition favors a new microbial configuration characterized by a more structured and complex bacterial community which, in turn, can be associated with enhanced functional expression within belowground communities [69,70].

### 3.2. Combination of Different Maize/Bean Lines in Intercropping Significantly Alters the Composition and Structure of the Rhizosphere Microbiome

Bacterial community richness in bean lines seems to confirm the hypothesis of a core microbiome [71] shared by different genotypes of *P. vulgaris*, while the significantly lower richness in *P. coccineus* (pC) ulteriorly validates the species-specific effect on microbiome recruitment. Despite not being significantly different, the mean of diversity values identifies two distinct groups determined by the two lines p83 and p91, showing lower bacterial richness and the two landraces pM and pMNE2, which show higher values. Maize lines also share a core microbiome [72] but the richness changes significantly in some IC combinations with maize lines O and BP, suggesting that they react to associated beans. Shifts in means of bacterial richness, even when not significant, between maize in SC compared to their IC combinations clusters separately for IC combinations with p83 and p91 and IC combinations with pM and pMNE2, showing lower and higher values, respectively. Interestingly, the observed shifts resemble the behavior of bean lines in SC, leading to the hypothesis that bean lines have an impact on the definition of the new microbial configuration in IC. 

Dissimilarity analysis [73] shows that BP and O experience a significantly consistent host-line dependent stratification of the microbiome determined by the genotypes involved. Moreover, it confirms a differential trend between IC combinations with p83 and p91 and those with pM and pMNE2 that cluster in space and are characterized by diverse bacterial community compositions. The genotype-dependent host effect is shaped by shifts in members of the phyla Proteobacteria, Bacteroidetes, Actinobacteria, Gemmatimonadetes and Acidobacteria, thus variations of many taxa, even if of limited entity, rather than by single ones influence the configuration of the IC microbiome in various combinations (Figure 4). This is in line with results obtained from Alpha-diversity analysis where the contribution of rare taxa determined the significant differences highlighted by Chao1 index. Further confirmations on the stronger impact of IC condition on the microbial community of the rhizosphere of O and BP maize genotypes derive from the numbers of differentially expressed bacterial genera between SC and IC conditions. Indeed, BP and O maize lines are the ones showing higher numbers of differentially expressed taxa in IC combinations compared to SC, with BP showing the majority of changes within the phyla Actinobacteria and O in the phyla Proteobacteria. Exploring more in detail the bacterial members highlighted for shifting their abundances from SC to IC condition, members of the Burkholderiaceae family appear recurrently downregulated in IC [31]. Burkholderia are involved in various mechanisms from N-fixation to plant growth promotion [74]. Li et al. [75] found that Burkholderiales decrease their abundance with N fertilizer application, suggesting that they may be inhibited by high soil N [76]. However, the majority of taxa that are recurrently reported for being differentially expressed between SC and IC appear to be consistently upregulated in the IC condition. O in IC shows recurrent taxa upregulated in IC with PGPB role or N fixation ability [77,78,79]. BP in IC shows several members of phylum Actinobacteria that are upregulated compared to SC, including Nocardiaceae and Frankiales which are known PGPR and nitrogen-fixing bacteria [65,80]. O shows more changes when intercropped to p83 and pM with enrichment of Proteobacteria members as Synthrophobacterales and Rhizobiales. S shows enrichment in Syntrophobacteraceae and Rhizobiales when intercropped with p83. Notably, the entire phyla Rokubacteria appears as differentially expressed only in BP and O. Overall, from differential analysis emerges that O and BP are more responsive to IC, followed by Spinato while I line seems the less responsive. Furthermore, certain bacterial members belonging to the phyla Proteobacteria and Actinobacteria appear to be particularly engaged in the IC condition as they are preferentially recruited by more responsive maize lines and could be regarded as potential candidates for assessing genotype combinations in the context of IC.

Co-occurrence networks enable the evaluation of the structure and functionality of bacterial communities [27,29,33,61,65,81,82,83,84] and evidence suggests that complex networks with a higher number of degrees and edges and lower modularity correspond to increased function expression in belowground communities [65]. In our study, co-occurrence networks were used to compare community structure among SC and IC combinations. Despite the impossibility of building networks for all conditions (due to the limited number of samples), extracted network properties and their comparison among conditions provide interesting cues. Again, two groups can be distinguished within bean lines with p91 showing lower network complexity compared to pM and pMNE2 and this dynamic is observed in IC combinations as well. Similar behavior seems shared by IC with p83, thus we may hypothesize that p83 possibly shares a similar community structure with p91. Hence, it is possible to deduce that the trend observed for bacterial diversity in IC combinations is mimed by the behavior of co-occurrence network complexity. Indeed, two distinct groups characterized by higher or lower network complexity are highlighted (Figure 6). Moreover, this dynamic seems again to be influenced by the bean line associated which determines the shifts in the community structure of the maize lines from SC condition towards the one of IC. Furthermore, co-occurrence network properties also confirm the pivotal role of Actinobacteria and Proteobacteria as they are generally more represented in all networks. 

## 4. Materials and Methods

### 4.1. Plant Materials and Experimental Design

This study was conducted at the farm of the Polytechnic University of Marche (UNIVPM) “P. Rosati” (43°33′47.8″ N 13°25′44.6″ E, Marche region, Italy) for two consecutive years 2018 and 2019 (climatic data can be found at http://www.meteo.marche.it/news/anno2018/clima2018.pdf and http://www.meteo.marche.it/news/2019/anno/clima2019.pdf for the year 2018 and 2019, respectively, accessed on 15 January 2020). The plant material selected for the experiment comprised four maize lines (*Zea mays* L.), which included three Italian landraces [85] (Spinato -S-, Ottofile -O-, Biancoperla -BP-) and one commercial hybrid (Italian Marano, flint type Ibrido -I-), and five bean lines. For maize lines, we used seeds directly supplied by local producers (custodian farmers) who multiply the seeds in accordance with regional seed production regulations (for Spinato see [86], for Biancoperla see [87], for Ottofile see [88]), while beans genetic material derived by Single Seed Descent -SSD- protocols [89]. The bean lines (*Phaseolus vulgaris* L.) consisted of one Italian (BEAN-ADAPT code EcE083, referred to as p83) and one Greek landrace (BEAN-ADAPT code EcE091, referred to as p91), recognized as high-yielding based on results from the BEAN-ADAPT Project [89,90] and two local varieties from Italy (BEAN-ADAPT code EcE136, referred to as pM) and Montenegro (referred to as pMNE2), with the latter traditionally grown in intercropping with maize in Montenegro). Additionally, one runner bean (*Phaseolus coccineus* L.) landrace from Italy (known as Clusven bean, referred to as pC) was included in the study (Table 3).

Each line was grown in sole crop (SC) and in full pairwise design for a total of twenty intercropping (IC) combinations (4 maize lines × 5 bean lines). Each condition was replicated three times in a complete randomized design. Each plot measured 3 m × 3.2 m with 1 m between replicates. Bean and maize lines were sowed on rows and in IC beans were planted after the emergence of maize, at about 5 cm from the maize stalk. Standard densities for maize (0.80 m between rows and 0.30 m among individuals in the same row) and bean (a plant every 0.30 m inside the row, 0.70 m between rows) were used. Maize BP was used as a border. In both years, three weeks before sowing, the field was weeded. No other treatment or fertilization was carried out during growing season, except ones for corn borer on calendar dates. Beans were sown two weeks after maize and one month later the field was weeded by hand and a pass between the rows was made with the rotary tiller. Throughout the growing season of the trial, relief irrigation was provided by means of a drip system. Bread wheat (*Triticum aestivum*) was used as the previous crop for both years of the trial. In bean monoculture plots, support nets were installed to allow the beans to climb (Appendix A). 

At flowering time, plant root systems were excavated and from each plant, approximately 15–20 cm below-ground, was sampled the soil gently brushed away from root tissues defined as the rhizosphere. Respectively, 54 and 41 rhizosphere samples were collected in the first and second year of field experiment for a total of 95 samples (Metadata table, Appendix A). Physical/chemical characteristics of the soil are reported in Appendix A. Soil analyses were performed by the ASSAM (Regional Agrochemical Centre, Jesi (AN), Italy) Analysis Laboratory using standard operational procedures. Samples were put in a tube and immediately stored in liquid nitrogen and successively at −80 °C. 

### 4.2. Metabarcoding Analysis

Genomic DNA was extracted from approximately 250 mg of each sample by means of the DNeasy PowerSoil kit (Qiagen, CD Genomics Company, Shirley, NY 11967, USA) following manufacturer instructions with slight modifications, i.e., time for cell lysis through vortexing at maximum speed was increased up to thirty minutes. Samples concentration was measured with a Qubit BR dsDNA assay kit (Thermofisher Thermo Fisher Scientific Inc., Waltham, MA, USA) and showed to be between 2 ng/μL and 120 ng/μL. Profile on a Tape Station 4120 with a genomic ScreenTape showed a DIN > 6 for the 8 samples randomly tested. DNA libraries were prepared with the Swift Amplicon 16S Panel kit (Swift Biosciences, Ann Arbor, MI, USA) according to protocol with 50 ng as starting material. Libraries were sequenced on a NovaSeq 6000 Illumina (San Diego, CA, USA) platform with a SP flow cell in a 250 PE mode and about 1 to 2 million fragments were produced per sample. The quality control (QC) was performed using Fastqc v 0.11.9; reads showed a very high average sequencing quality. The analysis on 16S region was performed on 600,000 reads (twice as much as recommended by Swift Bioscences). The analysis was carried out using the QIIME2 analysis workflow provided by Swift Biosciences and available at https://github.com/swiftbiosciences/q2_ITS, (accessed on 15 January 2021) with default parameters. Reads carrying 16S ranged from 380,568 to 588,712 reads, while reads passing the denoising pipeline to the classification process ranged from 327,910 to 498,128 reads. Reads from 16S were then classified using the Naïve–Bayes classifier trained on Silva_v132_99_16S. A total of 3134 species were identified. All statistical analyses were performed in R Software, R version 4.1.1 [91].

### 4.3. Statistical Analysis

Analysis of metabarcoding data was conducted focusing on two principal aims: first, we tried to determine the comprehensive trends experienced by the rhizosphere microbiome when plants are grown in intercropping (IC) conditions compared to sole crop (SC). Considering multiple lines enclosing part of the variability of a particular species involved in co-cultivation enables the detection of reliable trends and signals strictly associated with the IC condition. This part of the analysis pipeline, thus, focused on the comparison between maize in SC, bean in SC and IC (maize/bean) condition, where all lines of maize and all lines of beans concurred in the analysis without distinction between specific genotype combinations. Second, we tried to determine if the impact on the rhizosphere microbiome could be extended at the genotype level, beyond the one determined by the plant species involved. Shedding light on the potential effect of different genotype combinations on the recruitment of specific bacterial taxa may provide important cues for the efficient selection of specific lines for intercropping cultivation. For these analyses, each line (in SC) as well as each combination between all maize and bean lines (IC condition) employed in this study were considered separately. 

At this scope, richness and evenness diversity (Chao1 and Shannon) indexes were obtained with the QIIME2 pipeline on untrimmed data and multi-factorial anova along with Tukey HSD test was applied on obtained values (aov, base stats (v. 4.1.1)) to test the effect of “Cropping system” (SC vs. IC), “Plant status” (maize in SC vs. beans in SC vs. IC) and of “Accessions” within “Environment” (where “Environment” refers to a maize line in SC and its specific IC combinations, while “Accession” properly refers to all lines in SC and IC combinations). 

ASVs (Amplicon Sequence Variants) table was then filtered to concentrate on biologically relevant bacterial taxa using the phyloseq R package (v. 1.30.0) [92], obtaining a final table of 3134 taxa at species level. Venn diagram was utilized to characterize private bacterial members of the IC condition and Krona plot (KronaTools, v.2.7.1) was used to characterize the diversity of the IC subset through the percentage contribution of bacterial phyla and orders. Canonical analysis of principal coordinates (CAP analysis, [Partial] Distance-Based Redundancy Analysis, capscale function Vegan v.2.5) was performed on squared root transformed Bray–Curtis matrix [93] for each “Environment”. Anova-like permutation analysis was performed (anova.cca function [94]) to define the statistical significance of tested factors (“Accessions” within “Environment”). Constrained Principal Coordinate Analysis (CPCoA) is used to ordinate factor’s centroids and ASVs scores. Permanova was used to test the effects of “Plant status” and “Accession” within “Plant status” (strata() function). 

Linear Discriminant Analysis (LDA) Effect Size (LEfSe) [95] was used to identify biomarkers of plants in SC and IC conditions (LDA threshold 2.5, *p* value < 0.05). Differential abundance analysis between lines in SC and IC combinations was conducted with metagenomeSeq package [96] on a cumulative-sum scaled table (cumNorm function) and a Zero-Inflated Gaussian Distribution Mixture Model was applied (fitZig). Coefficients were used in a moderated *t*-test (makeContrasts and eBayes commands, Limma (v. 3.42.2) [97,98]) and *p* values adjusted with Benjamini–Hochberg (*p* value < 0.01). UpSetR (v.1.4) was used to concentrate on taxa differentially expressed (DE) proper of IC condition (taxa DE between beans and maize in SC and shared with taxa DE in beans in SC versus IC were excluded, Appendix A). 

Differential Network Analysis between maize and bean in SC compared to IC condition was conducted on Pearson correlation networks built with cluster_fast_gready function on the 900 most variable taxa (out of 2271 taxa). Fisher’s z-test was applied to identify differentially correlated (DC) taxa and *p* values adjusted with local false discovery rate (lfdr). Differential networks were plotted with the “union” function (NetCoMi). Properties and degree values of DC nodes were retrieved for each network. Co-occurrence networks were built for maize and bean lines in SC and for IC combinations with at least 5 replicates with the SparCC method (trans_network function, microeco (v.0.11) (https://github.com/stefpeschel/NetCoMi, accessed on 15 January 2021, [99]) on the normalized table (NetCoMi (v. 1.0.3)). The cor_optimization function found the optimal coefficient threshold and cor_cut value was set up for each network. Network properties were extracted with “Analyze network” tool in Cytoscape (v. 3.9.1).

## 5. Conclusions

The present study confirms that intercropping condition (IC) alters the rhizosphere bacterial population, but it underscores that the degree of this impact is substantially affected by the specific genotype combinations employed. Our data indicate that IC induces a novel microbial configuration marked by distinct bacterial species exclusive to this condition, supporting the hypothesis that crop diversity leads to novel exudates that alter the microbial community composition. Moreover, community structure in IC exhibits more intricate interconnections within bacterial communities compared to the sole cropping (SC) of maize. In particular, bacterial members recognized for their involvement in nitrogen fixation, such as the order Rhizobiales, significantly expand their network connectivity in IC as opposed to maize in SC. Notably, our findings emphasize that specific combinations of genotypes are significantly linked to shifts in the composition of bacterial communities. In particular, two distinct groups of IC combinations are highlighted for having higher or lower bacterial diversity and community complexity. The diversity of the rhizosphere microbiome appears to be influenced by the presence of associated beans in the IC system, and similar dynamics are observed in co-occurrence networks. In addition, maize lines in IC exhibit varying propensities in recruiting bacterial members, and among the taxa consistently enriched in IC, some may be promising candidates for evaluating genotype combinations in IC. In essence, our findings reveal that the presence of the associated bean line exerts a significant impact on shaping and altering the microbiome of maize when compared to its monoculture counterpart. Moreover, specific maize lines show different responses to the associated legume. Ultimately, our study not only suggests that genotypic selection could be important for adaptation to intercropping, but also that this must be carried out on both species employed. Furthermore, genotype combinations clearly affect the state of microbial communities in the rhizosphere, which could be evaluated to booster long-term benefits and enhance the sustainability of agricultural systems. We are convinced of the need for further studies to continue dissecting the role of the rhizosphere microbiome on the beneficial effects of different intercropping systems.

## Figures and Tables

**Figure 1 ijms-25-01288-f001:**
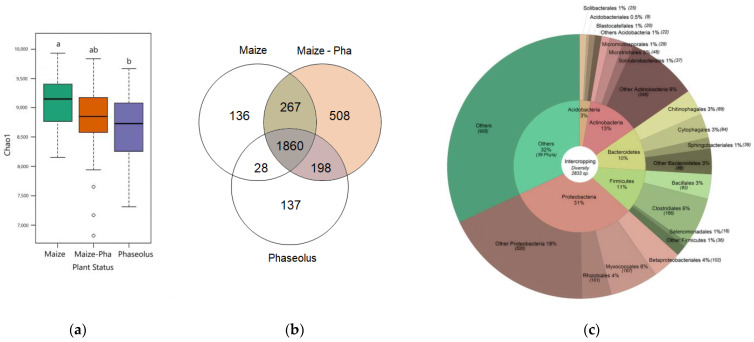
Bacterial richness in IC condition (**a**) Boxplots of Chao1 richness index values for means of maize and beans in SC and the IC condition (Plant Status). Letters above boxplot identify significant differences highlighted by Tukey HSD test. (**b**) Venn diagram of bacterial species belonging to each subset. (**c**) Doughnut plot of total bacterial diversity for the IC condition with percentages of phyla in the inner circle and of orders in the outer circle.

**Figure 2 ijms-25-01288-f002:**
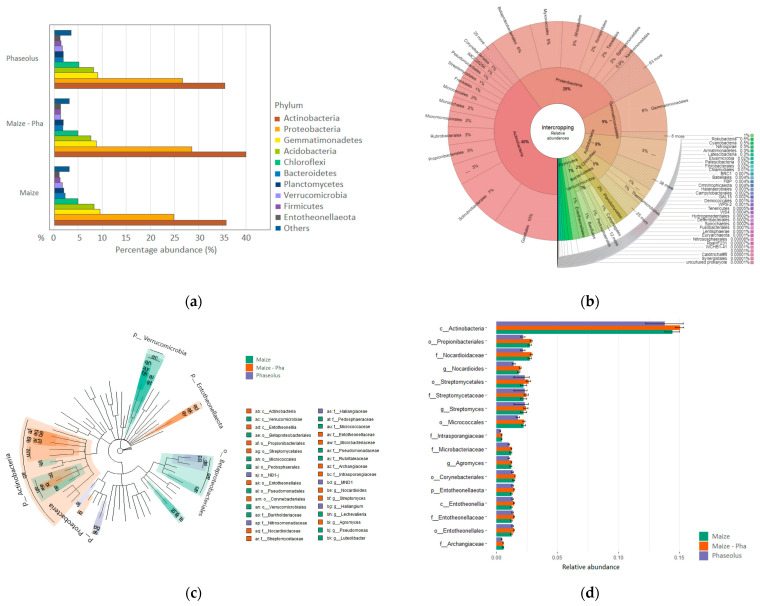
Bacterial community composition in IC condition (**a**) Barplot of relative abundances for Plant Status (Maize in SC, *Phaseolus* in SC, Maize–*Phaseolus* IC). (**b**) Krona plot of relative abundances of the IC condition. (**c**) Linear discriminant analysis Effect Size (LEfSe) at genus level based on Plant status. The plot shows enriched bacterial genera and their relative upper taxonomy assignment that resulted significantly associated with the three categories that are identified with different colors: Maize in SC (green), Phaseolus in SC (violet), Maize–Pha in IC (orange). (**d**) Barplot of relative abundances of bacterial members identified for being biomarkers of the IC condition.

**Figure 3 ijms-25-01288-f003:**
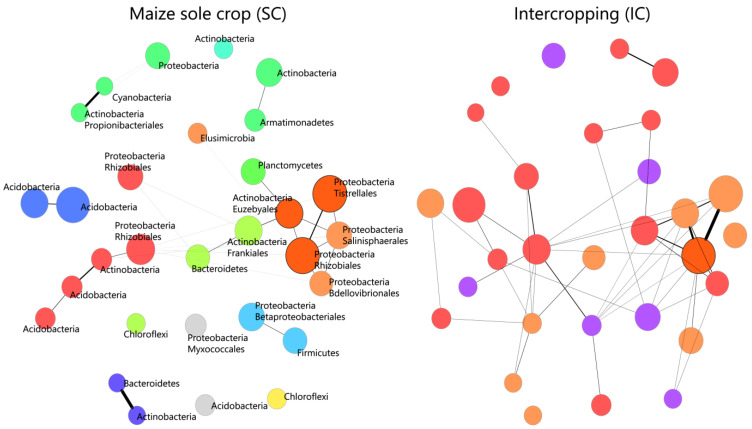
Differential correlation networks between maize in SC and IC condition. Correlation networks showing differentially associated taxa between maize in SC and the IC condition. Nodes and edges reported are the ones overpassing the lfdr threshold applied to Fisher’s z-test. Colors identify different clusters and thickness of lines identifies higher weight of the connection between the nodes. Phyla and/or orders of bacteria representing differentially expressed nodes are reported on the network of maize in SC as they correspond to the nodes of the IC network. Nodes with darker borders identify stronger differences in connections between the two networks. Node size is defined by higher clr values.

**Figure 4 ijms-25-01288-f004:**
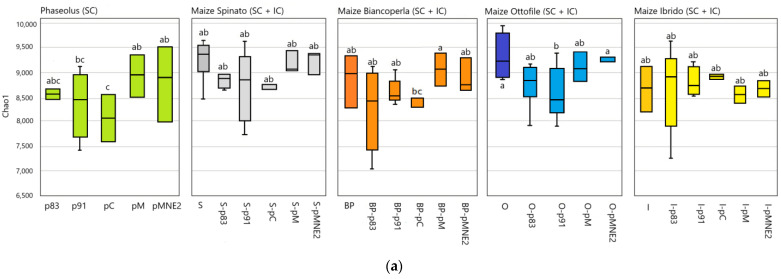
Bacterial diversity in different IC combinations. (**a**) Boxplots of Chao1 richness index values for means of each “Accession” of maize and beans grown in SC and for each maize/bean IC combination. Letters above boxplot identify significant differences highlighted by Tukey HSD test. (**b**) Constrained Principal Coordinate analysis (PcoA) and biplot of phyla scores of PcoA analysis based on Bray–Curtis distances of rhizosphere samples in IC systems of BP e O, constrained by “Accessions” including maize and beans in SC and different IC combination with the maize line. The percentage of variation explained by each axis refers to the fraction of the total variance of the data explained by the constrained factor (33% and 32% for BP and O systems, respectively). CAP analysis shows a significant effect on sample clustering by “Accessions” (*p* < 0.034 and *p* < 0.012 in BP and O systems, respectively). The arrows point to the centroid of the constrained factor. Circle size depicts the relative abundance of phyla (log scale) that contributed more in clustering samples. Colors illustrate different phyla as reported in the legend. Asterisks indicate different levels of statistical significance of the comparisons (*, *p*-value ≤ 0.05).

**Figure 5 ijms-25-01288-f005:**
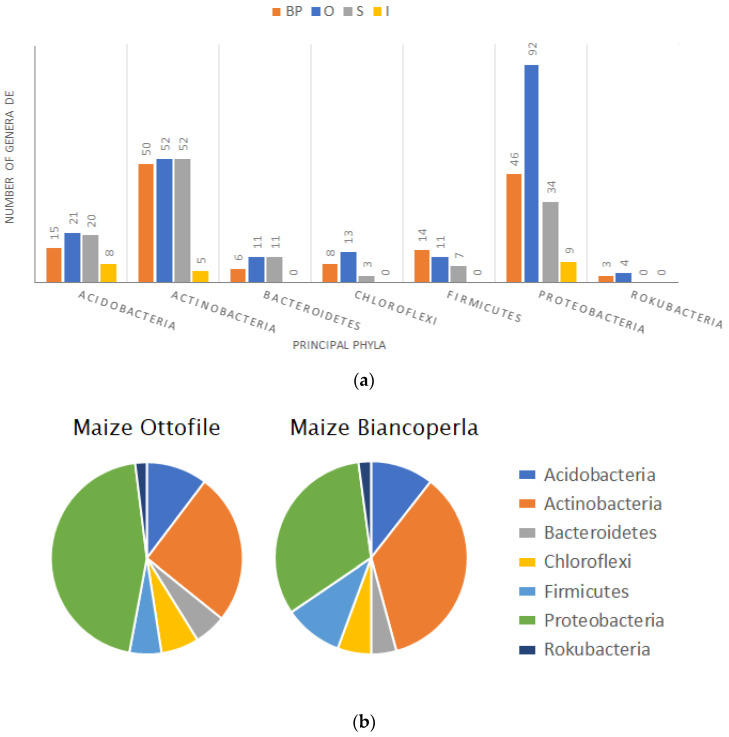
Bacterial genera differentially expressed in different IC combinations. (**a**) Composition of bacterial genera DE between maize in SC and relative IC combinations for each intercropping system (BP, O, S, I) grouped by their belonging to principal phyla identified. (**b**) Pie plots showing the phyla composition of total bacterial genera DE between BP and O in SC and respective IC combinations. (**c**) Summary of the recurrent bacterial genera found to be upregulated in different BP and O IC combinations compared to BP and O in SC.

**Figure 6 ijms-25-01288-f006:**
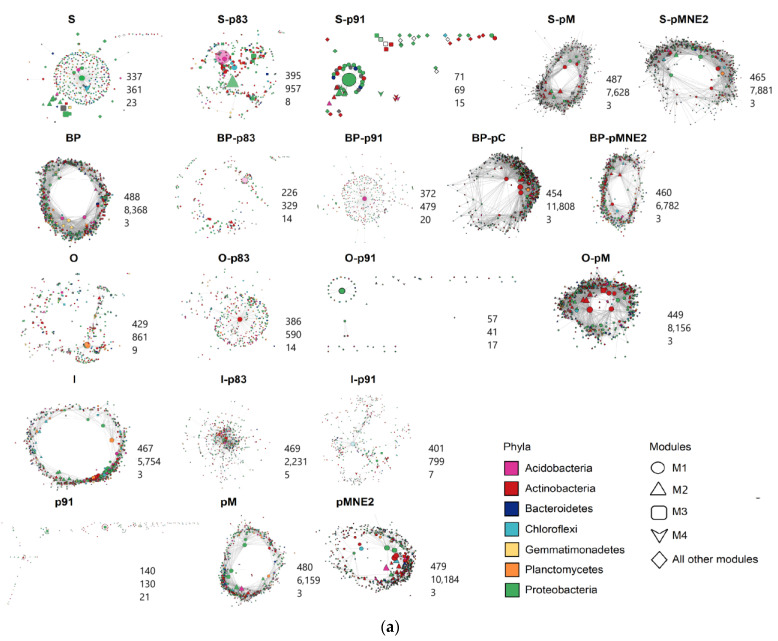
Co-occurrence bacterial networks for maize and bean in SC and relative IC combinations. (**a**) Co-occurrence networks based on SparCC method are reported for maize and beans in SC and for IC combinations and relative number of nodes, edges and modules are reported on the below-right corner of each network. Network representation was created using the Edge-weighted spring-embedded layout selecting “edge weight” as variable to build the network. Node shapes were defined based on modularity. Node size was defined using continuous mapping based on degree of each node. Node colors were defined by phyla (selection of more represented phyla with highest centrality values) and color transparency was defined on the basis of nodes betweenness centrality with lower values having more transparency. Edges were represented in gray with transparency based on continuous mapping of edge weight with low-weight edges being more transparent. Slight adjustments of values were defined for each network. (**b**) Nodes for each network are grouped at phylum level and represented with different colors. Number of nodes belonging to each phylum are reported below each graph. Variation in degree values of nodes (connections) grouped by phyla are reported on y axis for each network.

**Table 1 ijms-25-01288-t001:** Number of connections of nodes differentially correlated between maize in SC and IC. Nodes significantly differentially correlated between the correlation networks for maize in SC (M_SC) and for IC are reported and grouped for their belonging to bacterial phyla. Relative number of connections in both conditions is shown.

Phyla	Bacterial Genera	M_SC	IC
Acidobacteria	Holophagae_Subgroup7_uncultured bacterium gp7	0	9
	Subgroup5_uncultured *Acidobacterium* sp.	4	18
	Subgroup6_unculture microorganism	2	30
	Thermoanaerobaculia_Thermoanaerobaculiales_Thermoanaerobaculaceae_Sub10	2	20
	Acidimicrobiaa_Micotrichales_Ilumatobacteraceae_uncultured bacterium	8	44
Actinobacteria	Actinobacteria_Elev-16S-976_uncultured bacterium	1	2
	Actinobacteria_Frankiales_uncultured	9	59
	Actinobacteria_Micromonosporales_Micromonosporaceae_Actinorabdospora	2	17
	Actinobacteria_Propionibacteriales_Propionibacteraceae_Haloactinopolyspora	1	6
	Actinobacteria_Streptosporangiales_Thermomonosporaceae_Actinocoralia	2	11
	Nitriliruptoria_Euzebyales_Euzebyaceae_uncultured	12	80
	Thermoleophilia_Solitubrobacterales_67-14_uncultiured Rubrobacteraceae	3	17
Armatimonadetes	Uncultured bacterium_#0319-6E2	1	28
Bacteroidetes	Bacteroidia_Chitinophagales_Chitinophagaceae_Vibrionimonas	1	22
	Ignavibacteria_OPB56_uncultured bacterium_#0319-6E22	5	8
	Anaerolinae_R8G-13-54-9_uncultured bacterium	2	30
Chloroflexi	TK10_bacterium Ellin6543_bacterium Ellis6543	2	67
Cyanobacteria	Oxyphotobacteria_Chloroplast_Trifolium pratense	1	19
Elusimicrobia	Lineage 1b	0	11
Firmicutes	Bacill_Bacillales_Bacillaceae	3	14
Planctomycetes	VadinHA49_uncultured bacterium	2	12
Proteobacteria	Alphaproteobacteria_Rhizobiales_Beijerinckiaceae	20	97
	Alphaproteobacteria_Rhizobiales_KF-JG30-B3	20	81
	Alphaproteobacteria_Rhizobiales_Rhizobiales Incertae Sedis_Bauldia	4	27
	Alphaproteobacteria_Tistrellales_Germinicoccaceae_Candidatus Alysiosphaera	17	77
	Deltaproteobacteria_Bdellovibrionales_Bdellovibionaceae_Bdellovibrio	5	48
	Delaproteobacteria_Myxococcales_Bfdi19	1	1
	Deltaproteobacteria_Myxococcales_uncultered	0	40
	Gammaproteobacteria_Betaproteobacteriales_TRA3-20_uncultured Alcaligenaceae	3	31
	Gammaproteobacteria_Salinisphaerales_Solimonadaceae	8	11

**Table 2 ijms-25-01288-t002:** Permanova on Bray–Curtis index for different IC combinations. Permutational analysis of variance based on Bray–Curtis distance matrixes built for each intercropping system determined by maize lines. Asterisks indicate different levels of statistical significance of the comparisons (*, *p*-value ≤ 0.05; **, *p*-value ≤ 0.01).

Source	Df	Sum of Sqs	Mean Sqs	F Model	R2	Pr (>F)
Maize Spinato (S)						
Plant status	2	0.058	0.029	1.316	0.066	0.144
Accession	8	0.250	0.031	1.410	0.283	0.05 *
Residuals	26	0.577	0.022	0.651		
Total	36	0.886	1.00000			
Maize Ibrido (I)						
Plant status	2	0.072	0.036	1.516	0.088	0.084
Accession	8	0.249	0.031	1.305	0.303	0.116
Residuals	21	0.500	0.024	0.609		
Total	31	0.821	1			
Maize Ottofile (O)						
Plant status	2	0.048	0.024	1.242	0.061	0.216
Accession	7	0.255	0.036	1.864	0.322	0.004 **
Residuals	25	0.488	0.020	0.617		
Total	34	0.791	1			
Maize Biancoperla (BP)						
Plant status	2	0.045	0.023	1.061	0.054	0.357
Accession	9	0.290	0.032	1.503	0.341	0.02 *
Residuals	24	0.515	0.021	0.605		
Total	35	0.85	1			

**Table 3 ijms-25-01288-t003:** Plant material. For each genotype are reported the acronyms used in this study, the plant species, the description of the material (e.g., landraces), its geographical origin, the common name when available and the type of genetic material.

Acronyms	Species	Description	Origin	Common Name	Genetic Material
p83	*P. vulgaris*	Landrace	Italy	-	Single Seed Descent (SSD)
p91	*P. vulgaris*	Landrace	Greece	-	Single Seed Descent (SSD)
pC	*P. coccineus*	Landrace	Italy	Fagiolo di Clusven	Single Seed Descent (SSD)
pM	*P. vulgaris*	Landrace	Italy	Monachello	Single Seed Descent (SSD)
pMNE2	*P. vulgaris*	Landrace	Montenegro	-	Single Seed Descent (SSD)
S	*Zea mays*	Landrace	Italy	Spinato	Multiplied by custodian farmers
BP	*Zea mays*	Landrace	Italy	Biancoperla	Multiplied by custodian farmers
O	*Zea mays*	Landrace	Italy	Ottofile	Multiplied by custodian farmers
I	*Zea mays*	Commercial hybrid	-	Ibrido (tipo Marano)	Multiplied by custodian farmers

## Data Availability

The raw sequence reads generated and analysed during this study are available in the Sequence Read Archive (SRA) of the National Center for Biotechnology Information (NCBI) with the BioProject number PRJNA1030176.

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
