# Peer review of "Genotype Combinations Drive Variability in the Microbiome Configuration of the Rhizosphere of Maize/Bean Intercropping System"

_ijms, 2024, doi:10.3390/ijms25021288_

Round 1

Reviewer 1 Report

Comments and Suggestions for Authors

In this paper, the authors conducted a field study, probing the effects of intercropping and diverse maize and beans genotype combinations. The results will provide a theoretical basis for the study of plant-soil microorganism interactions. However, the results and analysis of the article require significant modifications and the addition of key soil physicochemical data

1. My biggest concern is the cause of changes in microbial structure and diversity in intercropping systems, which is definitely related to the physicochemical properties of rhizosphere soil, but the author did not measure the rhizosphere soil of different species.

2. The analysis of the results is very confusing. The experiment comprised four maize lines and five bean lines (Table 1). The article mainly focuses on different genotypes, and should analyze the structure and diversity of rhizosphere microorganisms under different combinations of varieties, for example, the network analysis in Figure 6.

3. Line103-105: Chao1 in figure 1(a) is an richness index, not diversity. You should give the Shannon index.

4. Line 462-492:The experimental design description is a bit confusing. 54 and 41 rhizosphere samples were collected at first and second year of field experiment for a total of 95 samples? What is the specific information about these 95 samples? Which combination of rhizosphere soil?

Reviewer 2 Report

Comments and Suggestions for Authors

Data and the methods for data analysis are missing from the abstract. There is nothing about the main policies implications.

Introduction presents in detail the main results obtained in previous studies related to cereals – legumes intercropping. However, the authors state only one main hypothesis - in addition to crop species, the choice of crop genotypes also influence the rhizosphere community’s composition (lines 71 to 74). It is not clear whether other factors like – initial microbial composition; type of soil; precipitations; etc have an influence on overall rhizosphere composition.

Material and methods section should follow the introduction part. The paragraph (between lines 81 and 98) cannot present in a comprehensive approach the research protocol and the methodology behind data analysis such as to properly understand the results section.

Result and discussion sections present in a comprehensive way the main findings of the paper. Policies implications are missing from the discussion section.

Reviewer 3 Report

Comments and Suggestions for Authors

Dear author, initially after analyzing your article with the anti-plagiarism program and getting 98% plagiarism, I wanted not to do this review. But the moment I opened and checked the sources I found that you have a preprint of this article which led to this problem. 

the same thing happened to me with an article I put in preprint and I had to do a lot of work to clear it up with one of the reviewers.

I will now give you my comments on your article:

1. in the abstract please, in the final part of the abstract, insert the following elements of the study (especially data)

2. the purpose of the article is presented vaguely, somewhere on line 84

3. the working technique/methodology is not presented (separate chapter) - was found at the end of the article (not where it belongs). please put it before the experimental results

4. The graphs obtained are very eloquent but no comment is made on the results obtained.

5. In chapter 3 (discussion) I don't think there is any purpose if there are no references to your experimental results (respectively to the figures in the article) and comparisons with the scientific literature.

6. since there are no comments on the results obtained, there is nothing contradictory in the conclusions either (they are general information). specific conclusions should be drawn from the study presented

Round 2

Reviewer 1 Report

Comments and Suggestions for Authors

The author did not carefully revise the four questions I previously raised. The experimental design, data, and result analysis are quite confusing.

The article mainly focuses on different genotypes, and should analyze the structure and diversity of rhizosphere microorganisms under different combinations of varieties.

Reviewer 2 Report

Comments and Suggestions for Authors The authors considered the main questions pointed out in the first review process. However intercropping and pesticide reductions should be discussed also from the other stakeholders point of view – farmers; society etc.

Reviewer 3 Report

Comments and Suggestions for Authors

I have only a few comments on the content of the article:

-please modify the contract in figure 3 to show the lines

- figure 4.b can be made larger

I agree with the publication of the article in its current form
